# The Association between Serum Testosterone and Hyperuricemia in Males

**DOI:** 10.3390/jcm11102743

**Published:** 2022-05-12

**Authors:** Meng-Ko Tsai, Kuang-Chen Hung, Chun-Cheng Liao, Lung-Fa Pan, Chia-Lien Hung, Deng-Ho Yang

**Affiliations:** 1Department of Internal Medicine, Taichung Armed Forces General Hospital, Taichung 41152, Taiwan; tsaimed141@gmail.com (M.-K.T.); lung-fa@803.org.tw (L.-F.P.); 2Division of Rheumatology/Immunology/Allergy, Department of Internal Medicine, Tri-Service General Hospital, National Defense Medical Center, Taipei 11490, Taiwan; 3Division of Neurosurgery, Department of Surgery, Taichung Army Force General Hospital, Taichung 41152, Taiwan; sur060@gmail.com; 4Department of Surgery, National Defense Medical Center, Taipei 11490, Taiwan; 5General Education Center, College of Humanities and General Education, Central Taiwan University of Science and Technology, Taichung 40601, Taiwan; 6Department of Nursing, College of Management, Central Taiwan University of Science and Technology, Taichung 40601, Taiwan; 7Department of Family Medicine, Taichung Armed Forces General Hospital, Taichung 41152, Taiwan; milkbottle97@yahoo.com.tw; 8School of Medicine, National Defense Medical Center, Taipei 11490, Taiwan; 9Department of Medical Education and Research, Taichung Armed Forces General Hospital, Taichung 41168, Taiwan; chia-lien@803.org.tw; 10Graduate Institute of Radiological Science, Central Taiwan University of Science and Technology, Taichung 40601, Taiwan; 11Department of Medical Laboratory Science and Biotechnology, Central Taiwan University of Science and Technology, Taichung 406053, Taiwan; 12Institute of Biomedical Science, National Chung-Hsing University, Taichung 40227, Taiwan

**Keywords:** testosterone, urate, gout, male, hyperuricemia, uric acid

## Abstract

Gout is a common systemic inflammatory disease with a male predominance. This study aimed to determine the relationship between serum total testosterone level and hyperuricemia. Data on 1899 men, collected from 2007 to 2017, were included in the analysis. Serum testosterone and urate (SU) were measured on enrolment. The primary endpoints were SU levels ≥ 7 mg/dL and ≥9 mg/dL. On enrolment, participants had a mean age of 45.6 years and mean total testosterone and SU levels of 510 ng/dL and 6.6 mg/dL, respectively. The mean total testosterone levels were 533 and 470 ng/dL in patients with SU levels < 7 mg/dL and ≥7 mg/dL, respectively (*p* < 0.001); and 515 and 425 ng/dL in patients with SU levels < 9 mg/dL and ≥9 mg/dL, respectively (*p* < 0.001). After adjusting for age, body mass index, creatinine, serum lipid, fasting blood glucose, systolic blood pressure, and diastolic blood pressure, low testosterone level (<400 ng/dL) was significantly associated with an SU level ≥ 7 mg/dL (hazard ratio: 1.182, 95% confidence interval: 1.005–1.39) and ≥9 mg/dL (hazard ratio: 1.905, 95% confidence interval: 1.239–2.928). In men, a low testosterone level may be associated with an increased risk of hyperuricemia.

## 1. Introduction

Gout is the most common inflammatory arthritis among adults, affecting healthcare costs, productivity, quality of life, and physical activity [1,2]. Medical conditions (such as metabolic syndrome and hyperuricemia), medications (such as diuretics), and biologic variables (such as age, genetic make-up, postmenopausal status in women, and male sex) are important risk factors for gout [2]. In Western countries, the prevalence is 1–2% in women and 3–6% in men. In men, the prevalence increases with age; but in women the prevalence increases with age only until menopause [2,3]. A study conducted in Taiwan between 1993 and 1996 found that the average serum urate (SU) level was 6.63 mg/dL for men and 5.62 mg/dL for women [4].

Gout is common in middle-aged men, but is relatively rare in women, especially in premenopausal women [5,6,7,8]. The determinants of the difference in the incidence of gout according to sex remain unclear. Although gout has been researched for decades, only a few studies have explored the reasons for the difference in incidence according to sex [9].

Some possible causes of the sex difference in gout have been proposed based on previous studies. In adults aged over 50 years, women have a longer life expectancy and higher prevalence of hypertension than men, therefore, gout may affect more older women [9]. In adults over 65 years, inappropriate prescribing has been reported to be more frequent in women than men [1]. A meta-analysis of studies on risk factors for gout according to sex did not identify any clear differences [10]. We hypothesized that gonadal hormones, especially total testosterone, may be associated with gout because of their effect on SU levels. Recent studies have shown varying results regarding the relationship between total testosterone and SU levels. Some studies have shown a positive association [11,12,13,14], and other studies have shown a negative association [15,16,17,18,19,20].

Previous studies were all cross-sectional, and no cohort studies have been used to estimate the effect of total testosterone on SU levels. Therefore, we conducted a prospective cohort study of adult men to determine the relationship between serum total testosterone and SU levels.

## 2. Methods

### 2.1. Study Participants

This prospective cohort study was conducted from 2007 to 2017 and used data from the MJ Health Screening Center, a private health examination institute in Taiwan. The data collected included patient demographics, high-density lipoprotein (HDL), low-density lipoprotein (LDL), total cholesterol (CHOL), triglycerides (TG), creatinine, estimated glomerular filtration rate (eGFR), fasting blood glucose (FBG), glycated hemoglobin (HbA1c), hemoglobin (Hgb), total testosterone, and SU. There were 128,705 patients in the database. After excluding females and males with missing data on SU at the enrolment visit, there were 7247 patients included. A further 5348 patients were excluded due to missing SU data during follow-up, leaving 1899 patients in the analysis (Figure 1). Comorbidities were defined according to clinical criteria. Hypertension was defined as systolic blood pressure (SBP) of 140 mmHg or diastolic blood pressure (DBP) of 90 mmHg. Hyperlipidemia was defined as CHOL level ≥ 200 mg/dL. Diabetes was defined as an FBG level ≥ 126 mg/dL.

### 2.2. Ethics Approval and Informed Consent

The study protocol was approved by the Institutional Review Board of the Tri-Service General Hospital (Study No. A202005160). The use of the data was authorized and the data were provided by the MJ Health Research Foundation (Authorization Code: MJHRF202103A). The requirement for informed consent was waived because the study used anonymized routinely collected data collected as part of routine health screening.

### 2.3. Assays of Lipids and Fasting Blood Glucose

A chemiluminescent microparticle immunoassay (Abbott ARCHITECT i2000) was used to measure total testosterone, a homogeneous direct method (TOSHIBA C8000) was used to measure HDL-cholesterol; the GPO-POD-ESPT method (TOSHIBA C8000) was used to measure TG, and the hexokinase/glucose-6-phosphate dehydrogenase method (TOSHIBA C8000) was used to measure FBG.

### 2.4. Serum Urate Levels

SU levels were dichotomized based on the risk of complications (such as, all-cause, stroke, cardiovascular mortality, and ischemic heart disease). Two separate cutpoints were used: 7 mg/dL and 9 mg/dL [21,22,23,24].

### 2.5. Definition of Low Testosterone

Hypogonadal symptoms in men aged <40 years can be associated with a total testosterone level of <400 ng/dL. As the participants were relatively young, we defined low testosterone as a total testosterone level <400 ng/dL [25].

### 2.6. Outcome Measurement

The primary outcome was a raised SU level at the end of the follow-up period. After the baseline assessment, all 1899 participants were followed up. After 10 years, the SU level was remeasured.

### 2.7. Statistical Analysis

Independent t-tests were used to assess the statistical significance of factor associated with the SU group (<7 mg/dL vs. ≥7 mg/dL; <9 mg/dL vs. ≥9 mg/dL). The mean values and standard deviations were calculated for age, HDL, LDL, CHOL, TG, creatinine, eGFR, FBG, HbA1c, and Hgb, and total testosterone was calculated according to the SU group. Cox regression was used to calculate unadjusted and adjusted hazard ratios (HRs) with 95% confidence intervals (CIs)to assess the association between the total testosterone level and the SU group (<7 mg/dL vs. ≥7 mg/dL and <9 mg/dL vs. ≥9 mg/dL). Participants were categorized according to their testosterone level (<400 ng/dL vs. ≥400 ng/dL). The Cox regression adjusted for age, BMI, CHOL, TG, creatinine level, DBP, SBP and FBG level. All statistical analyses were conducted using SPSS Version 22.0 (IBM Corp., Armonk, NY, USA), and *p*-values < 0.05 were considered statistically significant.

## 3. Results

The baseline characteristics of the 1899 study participants are shown in Table 1. There were 1215 (64%) and 1804 (95%) participants with an SU level < 7 mg/dL and <9 mg/dL, respectively. The mean follow-up time was 2.18 years. There were 684 and 95 patients in the SU ≥ 7 mg/dL and SU ≥ 9 mg/dL groups, respectively, with a mean age of 46.25 ± 11.22 years and 43.26 ± 10.84 years, respectively. All lipid parameters except for HDL were higher in the SU ≥ 7 mg/dL group than in the reference group. The creatinine levels were 1.08 mg/dL and 1.12 mg/dL; the FBG levels were 105.55 ± 19.59 and 105.74 ± 17.46 mg/dL; and the Hgb levels were 15.18 ± 1.04 mg/dL and 15.29 ± 1.06 mg/dL in the SU < 7 mg/dL and SU ≥ 7 mg/dL groups, respectively. The differences in the FBG and HbA1c levels between groups were not statistically significant. Age, cholesterol, HDL, LDL, triglycerides, and creatinine levels, and the EGFR differed significantly according to the SU group. Most of the findings, except for HDL and LDL levels, were similar when the data were analyzed using an SU cut-point of 9 mg/dL. The prevalence of hypertension and hyperlipidemia differed significantly according to the SU level, but the prevalence of diabetes did not differ significantly according to the SU level.

### 3.1. Association between the Serum Total Testosterone Level and the Serum Urate Level in Study Participants

The mean baseline SU levels of each group were 5.98 ± 1.05, 7.70 ± 1.15, 6.48 ± 1.25, and 8.85 ± 1.41 mg/dL in the SU < 7 mg/dL, SU ≥ 7 mg/dL, SU < 9 mg/dL, and SU ≥ 9 mg/dL groups, respectively. At the end of the study the mean SU level was 6.54 mg/dL, overall, and the SU levels were 5.74 ± 0.83, 7.97 ± 0.88, 6.37 ± 1.18, and 9.69 ± 0.66 mg/dL, in the SU < 7 mg/dL, SU ≥ 7 mg/dL, SU < 9 mg/dL, and SU ≥ 9 mg/dL groups, respectively. The SU level at the end of the study was higher than the initial level in the SU ≥ 7 mg/dL and SU ≥ 9 mg/dL groups. The mean total testosterone levels in each group were 533 ± 205, 470 ± 208, 515 ± 210 and 425 ± 143 ng/dL in the SU < 7 mg/dL, SU ≥ 7 mg/dL, SU < 9 mg/dL and SU ≥ 9 mg/dL groups, respectively. The total testosterone level differed significantly between the SU ≥ 7 mg/dL and SU ≥ 9 mg/dL and the respective reference groups.

### 3.2. Risk Factors for Developing Hyperuricemia ≥ 7 mg/dL

In the unadjusted Cox regression analysis, creatinine level was the variable with the strongest association with an SU level ≥7 mg/dL (crude HR: 1.88, 95% CI: 1.32–2.69) (Figure 2). A testosterone level < 400 ng/dL was the variable with the second strongest association with a SU ≥ 7 mg/dL (crude HR: 1.286, 95% CI: 1.10–1.50). After adjusting for age, BMI, creatinine, TG, CHOL, FBG, SBP, and DBP, the adjusted HR for a testosterone level < 400 ng/dL was 1.182 (95% CI: 1.005–1.39).

### 3.3. Risk Factors for Developing Hyperuricemia ≥ 9 mg/dL

In the unadjusted Cox regression analysis, creatinine level was also the variable with the strongest association with an SU level ≥ 9 mg/dL (crude HR: 3.902, 95% CI: 2.32–6.57) (Figure 3), and a testosterone level < 400 ng/dL was also the variable with the second strongest association with an SU level ≥ 9 mg/dL (crude HR: 2.176, 95% CI: 1.45–3.26). After adjusting for age, BMI, creatinine, TG, CHOL, FBG, SBP, and DBP, the adjusted HR for a testosterone level < 400 ng/dL was 1.905 (95% CI: 1.239–2.928).

### 3.4. Risk of Developing Hyperuricemia According to the Total Serun Cholesterol Level after Adjusting for Chronic Diseases

Participants with hyperuricemia had a higher prevalence of chronic disease, including hypertension and hyperlipidemia. After adjusting for age, BMI, creatinine, and comorbidities including hypertension, diabetes, hyper-lipidemia, participants with a testosterone level < 400 ng/dL (HR: 1.203; 95% CI: 1.024–1.414) had a higher risk of progressing to hyperuricemia (SU level: ≥7 mg/dL) than those with a testosterone level ≥ 400 ng/dL. Participants with a testosterone level <400 ng/dL also had a higher risk of progressing hyperuricemia (SU level: ≥9 mg/dL) than those with a testosterone level ≥ 400 ng/dL (HR: 2.024; 95% CI: 1.316–3.112). The adjusted HR of developing hyperuricemia is shown in Table 2. 

## 4. Discussion

This prospective cohort study evaluated the association between serum total testosterone levels and SU levels. Significant lower levels of serum total testosterone level were observed in men with hyperuricemia (SU level ≥ 7 mg/dL or SU level ≥ 9 mg/dL). Age, cholesterol, HDL, LDL, triglycerides, and creatinine levels, and the EGFR differed significantly according to the SU level group. Participants with hyperuricemia were significantly older and had a higher prevalence of hyperlipidemia, higher BMI, higher blood pressure, and a higher prevalence of renal insufficiency. Compared with the participants with a high serum testosterone (≥400 ng/dL), the participants with a low level of serum testosterone (<400 ng/dL) had a higher risk of progressing to hyperuricemia (SU level ≥ 7 mg/dL and SU level ≥ 9 mg/dL). Our study showed that a low testosterone level at enrolment was associated with an increased risk of having an elevated SU level, and developing an elevated SU level subsequently. Age and male sex are well-known risk factors for gout. Total testosterone levels decrease with age [26,27,28]. Therefore, it is not surprising that males with low testosterone levels had a higher risk of increased SU levels. Our study results are similar to those of a previous study on total testosterone levels. The normal range in serum testosterone in nonobese adult males aged 19 to 39 years in America and Europe is 264–916 ng/dL [29]. The SU levels among the participants in our study were higher than those reported in studies conducted in Western countries [21]. This may be due to genetic factors. The prevalence of hyperuricemia in Taiwan is higher than in other countries [30,31,32,33]. The SU levels in our study were similar to those found in a nutritional and health survey conducted in Taiwan from 1993 to 1996, which showed that the mean SU level in males was 6.63 mg/dL [4]. Several studies have assessed the relationship between the total testosterone and SU levels [11,12,13,14,15,16,17,18,20,34]. Han et al. [20] conducted a study among 7796 patients in China from 2011 to 2016 and showed that the SU levels were negatively associated with total testosterone level in adult males. The mean age of their study participants was 47.6 years, and the mean total testosterone level was 417 ng/dL, which are similar to those in our study. Their study showed that SU, BMI, and age were inversely associated with total testosterone levels.

Another study also showed that the total testosterone level was negatively associated with SU in diabetic patients. Wan et al. [35] conducted a cross-sectional study of 4426 diabetic patients in China in 2018. Participants with higher total testosterone level had a 48% lower risk of hyperuricemia. Feldman et al. [11] included 38 asymptomatic men with hyperuricemia with 31 men with normal SU levels as a control group, and they found that the serum total testosterone levels in the asymptomatic hyperuricemic group and the control group were 504.3 and 533.1 ng/dL, respectively. Our study was consistent with previous studies and strengthens the evidence of an association between total testosterone and SU levels.

The mechanisms underlying the association between the total testosterone and SU levels remain unclear, but several potential mechanisms have been identified. First, a low testosterone level may result in insulin resistance which has a negative correlation with SU clearance [35,36,37]. Second, testosterone has an effect on whole-body protein metabolism and can prevent protein loss [38]. Testosterone is associated with the activities of purine phosphoribosyl transferase with increasing purine production [39]. Third, sex hormones may primarily affect the SU level through its effect on renal urate excretion [13,40]. In summary, increased purine production, decreased renal excretion of urate, and insulin resistance may account for the relationship between low serum testosterone and high SU levels.

The 2020 American College of Rheumatology Guidelines state that pharmacologic treatment is conditionally effective in patients with asymptomatic hyperuricemia [41]. In our study, participants with a testosterone level < 400 ng/dL had a high risk of progressing hyperuricemia, especially those with a baseline SU level ≥ 9 mg/dL. Drug treatment should be considered in patients with an SU level ≥ 9 mg/dL and a total testosterone level < 400 ng/dL. However, further studies are needed to confirm these results.

Our study had several limitations: First, chronic diseases including diabetes, hypertension, and cardiovascular events are associated with hyperuricemia, and we did not exclude patients with these chronic diseases. However, testosterone plays a major role in the metabolism of glucose and lipids [42]. Second, we did not have data on medication use. Third, serum estradiol levels were not assessed in this study. Fourth, we did not evaluate lifestyle habits associated with an increased risk of hyperuricemia (such as alcohol consumption and diet). Previous studies have shown that patients with gout had a higher intake of energy, protein, and alcohol, and had higher serum levels of vitamin B12, C-reactive protein, TG, and SU [43]. In our study, hyperlipidemia and a high BMI were also associated with the higher risk for hyperuricemia. Dietary factors are associated with the development of hyperuricemia. However, the clinical data in our study, including higher levels of LDL, TG, CHOL, and BMI among participants with hyperuricemia support the importance of dietary factors in determining the SU level.

In conclusion, our 10-year prospective cohort study showed that in men, a lower testosterone level is associated with an increased risk of developing hyperuricemia. Pharmacologic treatment should be considered in patients with hyperuricemia (SU level ≥ 9 mg/dL) and total testosterone level < 400 ng/dL, because of the high risk of developing progressively higher SU levels. However, further studies are needed to confirm whether pharmacologic treatment is beneficial.

## Figures and Tables

**Figure 1 jcm-11-02743-f001:**
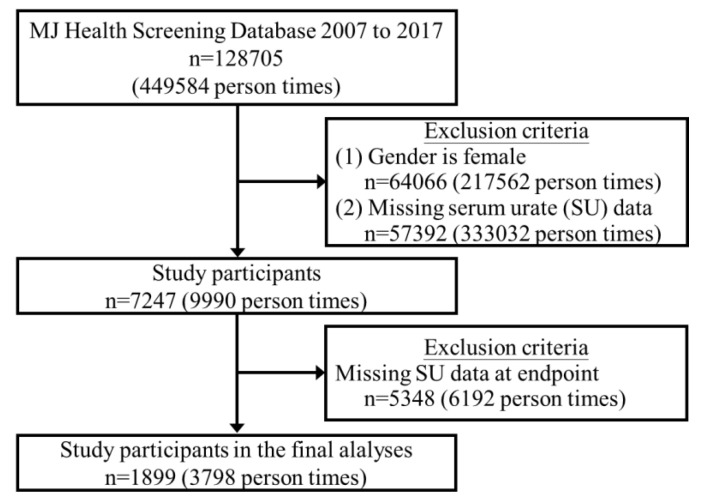
Flowchart of the sampling frame and participant section from the study database.

**Figure 2 jcm-11-02743-f002:**
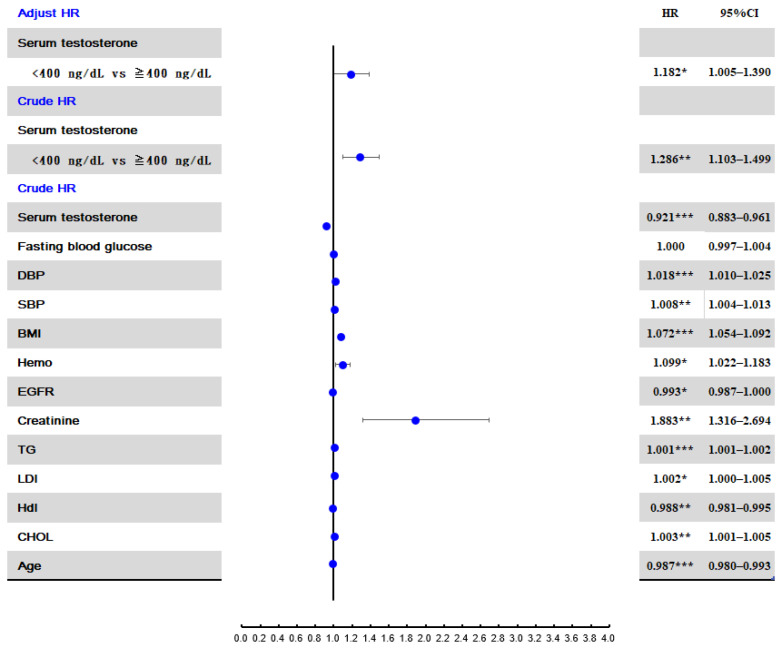
Risk factors for developing hyperuricemia (SU ≥ 7 mg/dL). Abbreviations: HR, hazard ratio; SU, serum urate. * *p* < 0.05; ** *p* < 0.01; *** *p* < 0.001.

**Figure 3 jcm-11-02743-f003:**
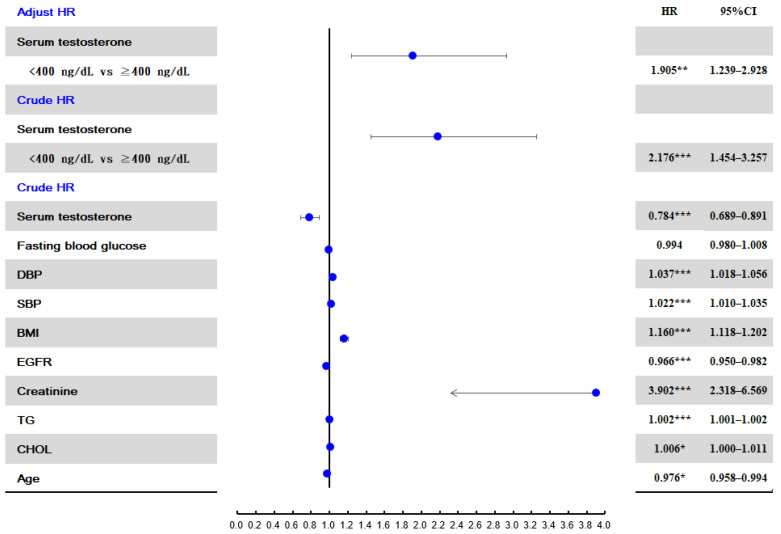
Risk factors for developing hyperuricemia (SU ≥ 9 mg/dL). * *p* < 0.05; ** *p* < 0.01; *** *p* < 0.001.

**Table 1 jcm-11-02743-t001:** Baseline participant characteristics according to their serum urate level.

	SU		*p*	SU		*p*
	<7 mg/dL	≥7 mg/dL		<9 mg/dL	≥9 mg/dL	
*n* (%)	1215 (64)	684 (36)		1804 (95)	95 (5)	
Age, year	46.25 ± 11.22	44.57 ± 11.22	0.002 **	45.77 ± 11.26	43.26 ± 10.84	0.034 **
BMI, kg/m^2^	24.33 ± 3.33	25.98 ± 3.77	<0.001 ***	24.77 ± 3.45	27.88 ± 4.60	<0.001 ***
Chol, mg/dL	197.16 ± 35.67	203.65 ± 35.45	<0.001 ***	199.07 ± 35.64	207.64 ± 36.45	0.022 *
HDL, mg/dL	52.23 ± 11.45	50.41 ± 10.80	0.001 **	51.64 ± 11.22	50.43 ± 11.81	0.310
LDL, mg/dL	121.14 ± 31.54	125.72 ± 33.44	0.003 **	122.51 ± 32.28	127.80 ± 32.48	0.120
TG, mg/dL	126.44 ± 102.66	164.84 ± 115.44	<0.001 ***	137.94 ± 108.39	184.56 ± 111.31	<0.001 ***
Creatinine, mg/dL	1.08 ± 0.16	1.12 ± 0.14	<0.001 ***	1.09 ± 0.15	1.18 ± 0.16	<0.001 ***
EGFR, mL/min/1.73 m^2^	80.63 ± 12.02	78.22 ± 13.07	<0.001 ***	80.06 ± 12.37	74.12 ± 12.76	<0.001 ***
SBP, mmHg	119.65 ± 14.80	123.65 ± 15.83	<0.001 ***	120.77 ± 15.24	127.41 ± 15.05	<0.001 ***
DBP, mmHg	76.67 ± 9.92	79.61 ± 10.84	<0.001 ***	77.51 ± 10.30	81.91 ± 10.69	<0.001 ***
FBG, mg/dL	105.55 ± 19.59	105.74 ± 17.46	0.832	105.71 ± 19.20	103.87 ± 10.09	0.355
HbA1C, %	5.33 ± 0.77	5.33 ± 0.68	0.978	5.33 ± 0.75	5.32 ± 0.47	0.940
Hgb, g/dL	15.18 ± 1.04	15.29 ± 1.06	0.028 *	15.21 ± 1.05	15.31 ± 1.09	0.348
Serum testosterone, ng/mL	5.33 ± 2.05	4.70 ± 2.08	<0.001 ***	5.15 ± 2.10	4.25 ± 1.43	<0.001 ***
Baseline SU, mg/dL	5.98 ± 1.05	7.70 ± 1.15	<0.001 ***	6.48 ± 1.25	8.85 ± 1.41	<0.001 ***
SU, mg/dL	5.74 ± 0.83	7.97 ± 0.88	<0.001 ***	6.37 ± 1.18	9.69 ± 0.66	<0.001 ***
Comorbidities ^a^						
Hypertension, *n* (%)	145 (12.6)	139 (21.3)	<0.001 ***	260 (15.2)	24 (26.7)	0.003 **
Diabetes, *n* (%)	90 (7.4)	40 (5.8)	0.196	127 (7.0)	3 (3.2)	0.144
Hyperlipidemia, *n* (%)	689 (56.7)	498 (72.8)	<0.001 ***	1114 (61.8)	73 (76.8)	0.003 **

BMI: body mass index; CHOL: cholesterol; DBP: diastolic blood pressure; EGFR: estimated glomerular filtration rate; FBG: fasting blood glucose; HbA1c: glycated hemoglobin; HDL: high-density lipoprotein; Hgb: Hemoglobin; LDL: low-density lipoprotein; SBP: systolic blood pressure; SU: serum urate; TG: triglycerides. Comparisons between SU group (<7 mg/dL vs. ≥7 mg/dL; <9 mg/dL vs. ≥9 mg/dL) were performed by the independent *t*-test. ^a^ The comorbidities of participants with and without hyperuricemia (<7 mg/dL vs. ≥7 mg/dL; <9 mg/dL vs. ≥9 mg/dL) were compared using a chi-square test. * *p* < 0.05; ** *p* < 0.01; *** *p* < 0.001.

**Table 2 jcm-11-02743-t002:** Risk of developing hyperuricemia according to the serum testosterone level and presence of comorbidities.

	SU ≥ 7 mg/dL		SU ≥ 9 mg/dL	
	HR	95% CI	HR	95% CI
Serum testosterone				
≥400 ng/mL	Ref.		Ref.	
<400 ng/mL	1.203 *	1.024–1.414	2.024 **	1.316–3.112
Comorbidities				
Hypertension	1.398 **	1.154–1.694	1.934 **	1.197–3.124
Hyperlipidemia	1.433 ***	1.202–1.708	1.488	0.911–2.430
Diabetes	0.840	0.600–1.175	0.446	0.139–1.435

The hazard ratios were adjusted for age, BMI, creatinine, and comorbidities including hypertension, hyperlipidemia, and diabetes. Abbreviations: BMI, body mass index; CI, confidence interval; HR, hazard ratio; SU, serum urate. * *p* < 0.05; ** *p* < 0.01; *** *p* < 0.001.

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
