# Peer review of "The Association between Serum Testosterone and Hyperuricemia in Males"

_jcm, 2022, doi:10.3390/jcm11102743_

Round 1

Reviewer 1 Report

This is an interesting research report about the association between serum testosterone and hyperuricemia in males.

 I have some remarks.

The mean values should always be given with a standard deviation.

Line 59-63. The authors’ arguments are not clear. I suggest describing this section in more detail.

Methods

In Methods, more accurate information about the analytical methods of all parameters should be given.

Line 75-79. This is not information about participants

Line 77 – total cholesterol

Line 95 – information repetition

Results

Figure 1 should be in the “Methods” section, and the qualification of participants to the research should also be moved to this section.

Only the most essential results from Table 1 should be repeated in the text.

The description of the results is unclear  and complicated

.Line 139 – “The mean baseline SUA level was 6.6 mg” this information is not in Table 1.

Table 1 does not need information about “p” under the table, but the statistical test used should be included. Value ranges should be added to the table.

Figures 2 and 3. The titles of figures should be more general. The interpretation should be in the text.

Discussion

Line 201-203 There is no information about endogenous ethanol in “Scovell et al.” work.

Author Response

This is an interesting research report about the association between serum testosterone and hyperuricemia in males. I have some remarks.

The mean values should always be given with a standard deviation.

Response:

We had corrected these descriptions of the mean values in this paper.

Line 59-63. The authors’ arguments are not clear. I suggest describing this section in more detail.

Response:

We had corrected as follow:

Some possible causes of gender difference in gout have been proposed from previ-ous studies. As women have a longer life expectancy and higher prevalence of hyperten-sion than men after 50 years, therefor, gout may affect more older women[9]. In adults over 65 years, inappropriate prescribing has been reported to be more frequent in women than men[1]. A meta-analysis assessed the risk factors for gout according to sex, did not identify any clear differences[10]. See line 59-64

In Methods, more accurate information about the analytical methods of all parameters should be given.

Response:

We had corrected the information as follow:

Independent t-test was applied for analyzing the SUA group (<7 mg/dL vs. ≥7 mg/dL; <9 mg/dL vs. ≥9 mg/dL). Mean values and standard deviation for age, HDL, LDL, CHOL, TG, creatinine, eGFR, FBS, HbA1c, Hgb, and total testosterone were calculated for SUA group. Unadjusted and adjusted hazard ratio (HR) with 95% confidence in-tervals (95% CI) was using Cox regression models to assess the association between total testosterone levels and the SUA group (<7 mg/dL vs. ≥7 mg/dL and <9 mg/dL vs. ≥9 mg/dL).  We categorized the participants into testosterone level (<400 ng/dL vs. ≥400 ng/dL).  The Cox regression model was applied for analyzing the association between the SUA group (<7 mg/dL vs. ≥7 mg/dL and <9 mg/dL vs. ≥9 mg/dL) after adjusting for the factors of age, CHOL, TG, creatinine levels, DBP, SBP and fasting blood glucose.  All statistical analyses were conducted using SPSS Version 22.0 software (IBM, Ar-monk, NY, USA), and a p-value <0.05, was considered statistically significant. See line 113-124.

Line 75-79. This is not information about participants

Response:

We had corrected as follow:

This prospective cohort study was conducted from 2007 to 2017, and used data from the MJ Health Screening Center, a private health examination institute in Taiwan. The data collected included patients’ demographics, high-density lipoprotein (HDL), low-density lipoprotein (LDL), total cholesterol (CHOL), triglycerides (TG), creatinine, estimated glomerular filtration rate (eGFR), fasting blood sugar (FBS), glycated hemoglo-bin (HbA1c), hemoglobin (Hgb), total testosterone, and SUA. There were 128,705 patients in the database. After excluding females and males with missing data on SUA at the en-rolment visit, there were 7,247 patients included. A further 5,348 patients were excluded due to missing SUA data during follow-up, leaving 1,899 patients in the analysis (Figure 1). See line 75-83.

Line 77 – total cholesterol

Response:

We had corrected.

Line 95 – information repetition

Response:

We had corrected. See line 102-103.

Figure 1 should be in the “Methods” section, and the qualification of participants to the research should also be moved to this section.

Response:

We had moved this description to the section of methods.

Only the most essential results from Table 1 should be repeated in the text. The description of the results is unclear and complicated.

Response:

We had added the description as follow:

This is a prospective cohort study to evaluate the association between serum total testosterone levels and SUA. Significant lower levels of serum total testosterone level in male with hyperuricema ( SUA ≥7 mg/dL or SUA ≥9 mg/dL ) were observed. Age, cho-lesterol, HDL, LDL, triglycerides, and creatinine levels, and the EGFR differed signifi-cantly according to the SUA group. Higher risk (HR: 1.182) of progressing hyperuricemia (UA ≧ 7 mg/dl) in the participants with low serum testosterone (< 400 ng/dl ) was ob-served when compared with the participants with high serum testosterone (≧ 400 ng/dl). Higher risk (HR: 1.905) of progressing hyperuricemia (UA ≧ 9 mg/dl) in the participants with low serum testosterone (< 400 ng/dl ) was observed when compared with the par-ticipants with high serum testosterone (≧ 400 ng/dl). Our study showed that low tes-tosterone levels at enrolment was associated with an increased the risk of having an ele-vated SUA, and of developing an elevated SUA subsequently. See line 179-190.

.Line 139 – “The mean baseline SUA level was 6.6 mg” this information is not in Table 1.

Response:

We had deleted this error description.  

Table 1 does not need information about “p” under the table, but the statistical test used should be included. Value ranges should be added to the table.

Response:

We had added the information as follow: Comparisons between SUA group (<7 mg/dL vs. ≥7 mg/dL; <9 mg/dL vs. ≥9 mg/dL) were performed by the Independent t-test. See Table 1.

Figures 2 and 3. The titles of figures should be more general. The interpretation should be in the text.

Response:

We had corrected Figure 2 as follow: Higher risk (HR: 1.182) of progressing hyperuricemia (UA ≧ 7 mg/dl) in the participants with low serum testosterone when compared with the participants with high serum testosterone.

We had corrected Figure 3 as follow: Higher risk (HR: 1.905) of progressing hyperuricemia (UA ≧ 9 mg/dl)  in the participants with low serum testosterone when compared with the participants with high serum testosterone.

Line 201-203 There is no information about endogenous ethanol in “Scovell et al.” work.

Response:

We had corrected as follow:

he mechanisms underlying the association between the total testosterone and SUA levels remain unclear, but several potential mechanisms have been identified. First, the secre-tion of total testosterone increases endogenous ethanol (EE) production and simultane-ously decreases SUA. Conversely, a higher SUA level inhibits the functional activity of the testes and lowers the level of EE[35]. See line 216-220.

Reviewer 2 Report

Thank you for giving me the opportunity to review this manuscript.

The manuscript is well written, the text is clear and easy to read.

This manuscript represents contribution to better understanding factors that determine dependence between serum testosterone and hyperuricemia in males, this topic is very interesting  but confirms previous knowledge from the literature.

The study presents the results of original research, includes a respectable follow-up duration of ten years but includes only one institution.

In the section Materials and methods authors clearly described the type of study and settings, participants, outcome measures and statistical analysis  but they did not take care about comorbidities with the most common chronic diseases, drugs and lifestyle. Authors defined comorbidities, drugs and lifestyle as limitations of the study but these impairs objectivity of the study.

In section Results authors clearly describe the sample which represents the remarkable size of the sample but they did not analyse comorbidities, drugs and lifestile and analyse the serum testosterone ans SUA for patients in only one Center.  

Conclusions are presented in an appropriate fashion and are supported by the data of the own research.

Manuscript covers interesting theme but sample and methodology are insufficient.

Author Response

Thank you for giving me the opportunity to review this manuscript.The manuscript is well written, the text is clear and easy to read.This manuscript represents contribution to better understanding factors that determine dependence between serum testosterone and hyperuricemia in males, this topic is very interesting but confirms previous knowledge from the literature. The study presents the results of original research, includes a respectable follow-up duration of ten years but includes only one institution.

Response:

The was a prospective cohort study from the MJ Health Research Foundation (Authorization Code: MJHRF202103A). The study used anonymized routinely collected data collected as part of routine health screening from different region of Taiwan. See line 88-91.

In the section Materials and methods authors clearly described the type of study and settings, participants, outcome measures and statistical analysis  but they did not take care about comorbidities with the most common chronic diseases, drugs and lifestyle. Authors defined comorbidities, drugs and lifestyle as limitations of the study but these impairs objectivity of the study. In section Results authors clearly describe the sample which represents the remarkable size of the sample but they did not analyse comorbidities, drugs and lifestile and analyse the serum testosterone ans SUA for patients in only one Center.

Response:

These people were relatively healthy in our study. Among the included people, 10.4% had a history of hypertension with medication; 4.4% had a history of diabetes with medication; 3.8% had a history of hyperlipidemia with medication. We had adjusted these confounding factors including blood pressure, glucose, TG, Chol, LDL to show lower testosterone levels are associated with increasing risk of hyperuricemia in adult males in a 10-year prospective study. See line 88-91.

Conclusions are presented in an appropriate fashion and are supported by the data of the own research.

Response:

In conclusion, our study showed that lower testosterone levels are associated with increasing risk of hyperuricemia in adult males in a 10-year prospective study. Pharma-cologic treatment should be considered in patients with hyperuricemia (SUA ≥9 mg/dL) and total testosterone <400 ng/dL, because of the high risk of developing higher serum uric levels. However, further studies are needed to confirm whether pharmacologic treatment is beneficial. See line 245-250.

Reviewer 3 Report

This work examines the association between serum uric acid levels and serum testosterone levels to better understand “pathogenetic mechanisms and Clinical Advances”. The experimental design, the selection of patients and, perhaps, the number of patients could be acceptable, as the stratification of uric acid levels and risk correlations are of some interest. Unfortunately, however, it has major limitations, as the authors also acknowledge:

patients with chronic diseases including diabetes, hypertension and cardiovascular events that could be associated with hyperuricemia are not excluded from the selection,

the medications taken by the patients were not considered,

the habits about increasing risk of hyperuricemia (such as alcohol consumption and mainly the diet*) were not taken into account.

Above all, there seems to be a lack of real novelty compared with the most recent studies, which, with a much larger number of cases, arrive at virtually the same conclusions.

This study should be completely  revised, perhaps even enriched with new data, in order to better highlight the new elements compared to the works already present in the literature.

*Dietary factors : Perim Fatma Türker , Mustafa Hoca , GülÅŸen Özduran .“The correlation of uric acid levels with biochemical parameters and dietary factors in individuals with asymptomatic hyperuricemia and gouty arthritis”. Nucleosides Nucleotides Nucleic Acids. . 2022 Mar 18;1-19. doi: 10.1080/15257770.2022.2051047

Typos:

Hyperuricecima => Line 137

Hyperuricmeia => Line150 and 160

Author Response

This work examines the association between serum uric acid levels and serum testosterone levels to better understand “pathogenetic mechanisms and Clinical Advances”. The experimental design, the selection of patients and, perhaps, the number of patients could be acceptable, as the stratification of uric acid levels and risk correlations are of some interest. Unfortunately, however, it has major limitations, as the authors also acknowledge: patients with chronic diseases including diabetes, hypertension and cardiovascular events that could be associated with hyperuricemia are not excluded from the selection, the medications taken by the patients were not considered, the habits about increasing risk of hyperuricemia (such as alcohol consumption and mainly the diet*) were not taken into account.

Response:

The was a prospective cohort study from the MJ Health Research Foundation (Authorization Code: MJHRF202103A). The study used anonymized routinely collected data collected as part of routine health screening from different region of Taiwan. These people were relatively healthy in our study. Among the included people, 10.4% had a history of hypertension with medication; 4.4% had a history of diabetes with medication; 3.8% had a history of hyperlipidemia with medication. We had adjusted these confounding factors including blood pressure, glucose, TG, Chol, LDL to show lower testosterone levels are associated with increasing risk of hyperuricemia in adult males in a 10-year prospective study. See line 75-91.

Above all, there seems to be a lack of real novelty compared with the most recent studies, which, with a much larger number of cases, arrive at virtually the same conclusions.

Response:

Several studies have assessed the relationship between total testosterone and SUA[11-18, 20, 34] . Han et al. conducted a study among 7,796 patients in China from 2011 to 2016 and showed that the SUA level was negatively associated with total testosterone in adult males[20]. The mean age of his study participants was 47.6 years, and the mean total tes-tosterone level was 417 ng/dL, which are similar to those in our study. Their study showed that SUA, BMI, and age were inversely associated with total testosterone[19]. Another study also showed that total testosterone was negatively associated with SUA in diabetic patients. Wan et al. conducted a cross-sectional study of 4,426 diabetic patients in China in 2018. Participants with higher total testosterone had a 48% lower risk of hyperuricemia[20]. Our study was consistent with previous studies and strengthens the evidence of an association between total testosterone and SUA. However, our study differs from that of a previous study. Feldman et al. included 38 asymptomatic hyperu-ricemic males and 31 normouricemic males as a control group, and they found that the serum total testosterone levels in the asymptomatic hyperuricemic group and the control group were 504.3 and 533.1 ng/dL, respectively[11]. See line 200-215

This study should be completely  revised, perhaps even enriched with new data, in order to better highlight the new elements compared to the works already present in the literature.

Response:

We had added description as follow:

From previous study, the patients with gout had higher intakes of energy, protein and alcohol consumption, and higher serum levels of vitamin B12, C-reactive protein, TG, and SUA was observed among them [41]. In our study, hyperlipidemia and high BMI were also associated with the higher risk for hyperuricemia. Dietary factors are associated with the development of hyperuricemia. The evaluation of diet was not done in our study. However, our study supported the importance of diet from the clinical data including higher levels of LDL, TG, CHOL and BMI. See 238-244.

Typos:

Hyperuricecima => Line 137

Response:

We had corrected.

Hyperuricmeia => Line150 and 160

Response:

We had corrected.

Round 2

Reviewer 3 Report

Thus revised, although it remains of reduced novelty, it becomes acceptable.